# Characterization of Bioactive Compounds and Novel Proteins Derived from Promising Source *Citrullus colocynthis* along with In-Vitro and In-Vivo Activities

**DOI:** 10.3390/molecules28041743

**Published:** 2023-02-11

**Authors:** Muhammad Afzal, Anis Shahzad Khan, Basit Zeshan, Muhammad Riaz, Umer Ejaz, Ayesha Saleem, Rida Zaineb, Haseeb Akram Sindhu, Chan Yean Yean, Naveed Ahmed

**Affiliations:** 1Department of Basic and Applied Chemistry, Faculty of Science and Technology, University of Central Punjab, Lahore 54000, Pakistan; 2Faculty of Sustainable Agriculture, Universiti Malaysia Sabah (UMS), Sandakan 90509, Sabah, Malaysia; 3Department of Medical Microbiology and Parasitology, School of Medical Sciences, Universiti Sains Malaysia, Kubang Kerian 16150, Kelantan, Malaysia

**Keywords:** *Citrullus colocynthis*, histopathology, secondary metabolites, antioxidant, antidiabetic, in silico analysis, antimicrobial protein

## Abstract

Herbal products are preferable to synthetic medicines, and the use of traditional medicines is increasing day-by-day. The current study was designed to evaluate the potentials of bioactive compounds from *Citrullus colocynthis* by performing FTIR, HPLC, and GC-MS analyses, which explore the good concentration of the secondary metabolites, such as gallic acid (74.854 ppm), vanillic acid (122.616 ppm), and ferulic acid (101.045 ppm) with considerable bioactivities. Antimicrobial protein was estimated by performing SDS-PAGE, ranging from 15 to 70 kDa in all protein fractions. The current study also checked the cytotoxicity of the bioactive compounds in the active fraction of *C. colocynthis*, and to perform this activity, the groups of rats were arranged with 16 rats randomly divided into four groups (three experimental and one control) by administering various dosage of methanolic fractions in dose-dependent manner. Histopathology was conducted on the livers of the rats after 15 days of sacrifice under deep anesthesia. In liver cell slides examined at the maximum dose of 600 mg/kg, minimal morphological changes, such as slight ballooning, nuclear variation, vacuolar degeneration, and hydropic degeneration, were observed. Furthermore, the in silico analysis identified bioactive compounds as potential drug candidates.

## 1. Introduction

Phytomedicine has remained popular across the globe as a treatment for a wide range of medical problems, due to its low cost and folkloric value. Holistic plant extract research is still thriving, as is the search for potent lead molecules from such extracts [1]. *Citrullus colocynthis* (*C. colocynthis*) is a traditional herb that grows in the south, center, and east areas of Iran to cure diabetes and inflammation [2]. The medication from *C. colocynthis* was found in the dried pulp and seeds, and it was used to treat hepatic and intestinal stimulants [3]. Asthma, gout, sciatica, leprosy, rheumatism, and paralysis [4], as an abortifacients, were among the conditions for which the medicine was effective [5]. Extracts of *C. colocynthis* possess a significant amount of phytochemicals, such as polyphenols, that act as antioxidants and scavengers of diverse reactive oxygen species, such as hydroxyl radicals and peroxy-free radicals [6]. Several studies have revealed the antioxidant potential of *C. colocynthis* fruit occurs due to presence of polyphenol content [7]. Among the endocrine diseases that affect carbohydrate metabolism, diabetes mellitus is the most common one [8]. The disease is associated with both microvascular (retinopathy, neuropathy, and nephropathy) and macrovascular (heart attack, stroke, and peripheral vascular disease) effects, which caused significant adversity and corporality [9]. *C. colocynthis* has shown bioactivity against diabetes, hypolipidemia, neoplasia, antioxidant, antibacterial, anti-inflammatory, pesticide, analgesic, and immunostimulant [10]. The most visible activity is fighting against diabetes. It is also considered a recommended seizure therapy. Diabetes, cancer, edema, and bacterial infections are commonly treated with the dried pulp of this plant [11].They have antioxidant properties, as well as effects on the liver and kidneys. Studies have revealed that leaves can aid in the treatment of diabetes. The useful treatment provided by the drug is for asthma, gout, sciatica, leprosy, rheumatism, and paralysis, as well as an abortifacient [12].

In addition to generating arginine, citrulline also plays a significant role in metabolic regulation. The amino acid arginine enhances wound healing, prevents cancer development early on, improves the functions of the heart, lungs, kidneys, digestive system, and immune system, in addition to restoring endothelial function.

In order to analyze the peptidome of *C. colocynthis*, MALDI-TOF mass spectrometry was used using chemical modifications of cysteine residues. Among the 23 cysteine-rich peptides in this plant, eight newly synthesized peptides, whose molecular weights range from 3650 to 4160 Da, have been purified using reverse-phase HPLC. A de novo assignment of proteolytic fragments of the b and y ions series determined their amino acid sequences.

Some studies show that it may bring about some aftereffects on liver cells. The comparison study of Herbs has been linked to drug-induced liver damage (DILI). The harm is caused by self-medication and the careless usage of natural medications. It is well-recognized that eating too much *Citrullus* pulp can be toxic. The heart, kidneys, liver, and female reproductive system are all known to be negatively impacted by *Citrullus*. After consuming *Citrullus*, cases of rectorrhagia, colitis, and bloody diarrhea have also previously been reported in the previous literature [13]. Regarding *Citrullus*’ hepatoprotective and hepatotoxic properties, there is a serious conflict of interest in the toxicity of a certain quantity of dose of *C. colocynthis*. The toxicity of the dosage of *C. colocynthis* is directly proportional to the increase in dose, as the liver is a delicate organ and most of the toxins are brought together to induce damage to the liver cells. Therefore, this study aimed to evaluate the histopathological changes in liver cells after giving a dose of *C. colocynthis* in white albino rats. With the help of GC-MS and HPLC, techniques were also confirmed by bioactive compounds present in the organic extract of *C. colocynthis*. Different solvent extracts made from *C. colocynthis*, including methanol, hexane, n-butanol, and chloroform, showed a positive result for some of the phytochemicals, such as terpenoids, flavonoids, and alkaloids [14]. According to studies, phytochemicals had different properties of bioactivities, including antioxidant, anti-diabetic, hemolytic, and thrombolytic activities [15], role in modulating the detoxicated enzymes, anticancer effect, and modulation of hormonal metabolisms. The current study was designed to explore the biopotential of valuable biological active phytoconstituents and proteins from the medicinal plant *C. colocynthis* to cure various infections. Because bacterial resistance is going on, increasing day-by-day due to overuse and misuse of drugs, it is time that a new source be explored, in order to find the secondary metabolites from medicinal plants that can be used as candidate drug molecules.

## 2. Results

### 2.1. GC-MS Analysis

GC-MS analysis of crude methanolic extract showed different bioactive compounds in crude methanol extract of *C. colocythis* (Figure 1). The prevailing compounds were ether, 3-butenyl pentyl oxime-, 1-octadecenoic acid methyl ester, and methyl stearate (Table 1).

### 2.2. HPLC Chromatogram of Methanolic Extract

As a result of the HPLC analysis of the *C. colocynthis* methanolic extract, seven phenolic compounds were detected, including quercetin, gallic acid, caffeic acid, benzoic acid, syringic acid, *p*-coumeric acid, and sinapic acid. Benzoic acid was the highest concentration (3.13283 ppm), followed by syringic acid (2.985425 ppm), as shown in Table 2.

### 2.3. FTR-ATR Analysis

FT-IR analysis was determined to further confirm the presence of functional groups to support the results of HPLC. The FT-IR spectra of methanolic extract contains the presence of hydroxyl (OH), cyclo alkane, phenol, C-O-C, C-H, C-S, and C-Br as functional groups. The results indicated the presence of phenolic acids and flavonoids in Figure 2 and Figure 3 and Table 3.

### 2.4. Antioxidant Activity of Methanolic Extracts

The results of antioxidant assay are shown in Table 4, in which 4 mg/mL of n-hexane fraction produced high DPPH scavenging activity among all the experimental plant extracts and fractions with the highest DPPH scavenging activity (52%), followed by 0.9 mg/mL of methanol extract (40%), chloroform fractions (34%), and n-butanol fraction (26%), whereas the least DPPH scavenging activity was observed in 0.9 mg/mL. Similarly, the minimum concentration, 0.15 mg/mL, of methanol extract produced high DPPH scavenging activity (28%), followed by 0.15 mg/mL chloroform fraction (26%), n-hexane fraction (25%), and n-butanol fraction (17%). Thus, it was observed that, by increasing the extract and fraction concentration from 0.15 mg/mL to 0.9 mg/mL, the % DPPH scavenging activity will be increased.

### 2.5. Thrombolytic Activity

The thrombolytic activity of the *C. colocynthis* was observed against human red blood cells by using different organic methanol extract (fraction of n-hexane, chloroform, and n-butanol). The methanol extract showed 4.5 ± 0.41 clot lysis, fraction of n-Hexane 5.5 ± 0.40, chloroform fraction 3.4 ± 0.37, and n-butanol fraction showed 7.5 ± 0.40 clot lysis (Figure 4). The results of anti-diabatic activity are shown in Table 5.

### 2.6. Hemolytic Activity

The results of hemolytic activity assay are shown in Table 6.

### 2.7. Histopathological Examination

After slaughtering the rats under deep anesthesia, for 48 h, preserve the liver section in 10% formalin and apply the standard micro techniques. The ready section of the liver slice was then stained with hematoxylin and eosin, and they were mounted in Canada balsam. A micrograph was produced then using light microscopy, which shows the results described in Figure 5, Figure 6 and Figure 7.

In the total protein quantification of all the three concentrations of protein, the samples were then subjected to sodium dodecyl sulphate polyacrylamide (SDS-PAGE) gel electrophoresis to find out the molecular weight of the proteins present in sample (Figure 8). The protein concentration of the extracts obtained from the Bradford assay was 600 µg/mL for the crude sample, 400 µg/mL for 70% fraction, and 500 µg/mL for 50% fraction. The ladder used for the reference was the page ruler pertained protein ladder, ranging from ~10 to ~180 kDa. Obvious bands were seen between ~20 and ~100 kDa in the 70% fractions, while in the 50% and crude, some bands were observed between ~10 to ~35 kDa and at 100 kDa.

The total protein quantification was determined from fruit of *C. colocynthis* through Bradford method. The results revealed that crude extract contain higher protein concentration (600 µg/mL ± 0.010) than the concentrations made, i.e., 50% and 70%. If we make a comparison between the concentrations, the 50% concentration contains a higher amount of protein (500 µg/mL ± 0.015), as compared to the 70% concentration (400 µg/mL ± 0.003). The previous results have been reported and revealed that fruit of C.C contain a noticeable amount of crude and true proteins, i.e., 19.16% and 10.8%, respectively [16].

Likewise, when we cross-examined the molecular weight of the proteins obtained from the ripen fruit of *C. colocynthis* through SDS-PAGE, the results revealed small-sized peptides ranging from ~70 and ~55 kDa, ~40 to ~35 kDa, and ~15 to ~25 kDa in crude, 70%, and 50% fractions. It was shown that different proteins were extracted from the seeds of the plant on the basis of MW and showed sizes of 150, 67, and 36 kDa [17]. Hence, we can conclude from the previous and current study that there are dynamic proteins present in the whole and different parts of the fruit of *C. colocynthis*.

### 2.8. Anti-Bacterial Activity

For anti-bacterial activity, the disc diffusion method was used against the Gram-positive and Gram-negative bacterial strains used for identifying the activity of peptides derived from the ripened fruit of *C. colocynthis* at different concentrations, and the zones of inhibitions were measured in mm, which is presented in Table 7. The results revealed vibrant anti-bacterial activity against Gram-positive and Gram-negative bacteria by comparing with antibiotics (imipenem, vancomycin) as a positive control, shown in Table 7.

Fractions of 50% and 70% revealed visible zones of inhibition, i.e., (8.83 ± 0.022) and (6.96 ± 0.047) against *Staphylococcus aureus*, (10.12 ± 0.017) and (8.58 ± 0.012) *Enterococcus faecalis*, (8.27 ± 0.001) and (5.86 ± 0.178) *Klebsiella pneumoniae*, and (6.4 ± 0.279) and (4.94 ± 0.473) against resistant strain of *Pseudomonas aeruginosa*. In contrast to the fractions, the crude sample exhibited an insignificant zone of inhibition against *Enterococcus faecalis* (2.76 ± 0.009) and *Staphylococcus aureus* (3.0 ± 0.001), while it showed noticeable a zone of inhibition (7.1 ± 0.178) against *Klebsiella pneumoniae*.

### 2.9. Molecular Docking

Docking determines the binding energies, and the highest Methyl stearate show high binding energy scores in free radical myeloperoxidase, NADPH-oxidase, and standard melatonin, while the trans-methyltrans-9-(2-butylcyclopentyl) nonanoate shows the second highest energy in the free radical myeloperoxidase, NADPH-oxidase, as compared to standard melatonin, which was shown in Table 8.

### 2.10. Assessment of Pharmacokinetics Properties

Swiss ADME analysis shows that methyl stearate is the most effective antioxidant compound against free radicals, since it has reasonable lipophilicity (4.81) and moderate water solubility. According to its physiochemical properties, it has a molecular formula of C_19_H_38_O_2_ and a molecular weight of 354.31 g/mol. According to the pharmacokinetic analysis, there is no violation of Lipinski’s rule. CYP2C9 and CYP2C19 were not inhibited by compound interaction and pharmacokinetic analysis. A boiled egg image of methyl stearate is shown in Figure 9.

## 3. Discussion

Medicinal plants are recommended to cure different ailments, due to their high potential and therapeutic properties [18]. Medicinal plants are used in traditional medicine to treat diabetes mellitus. *C. colocynthis* fruit has been identified as one of the most widely utilized sources in ethnobotanical investigations; hence, diabetic patients are advised to take citrullus fruits. Patients with diabetes are advised to drink infusions made from the fruits of the cucurbitaceous plant. Certain antidiabetic plants also include polysaccharides, gums, and glycans, in addition to alkaloids and polyphenols [19]. The presence of these phytoconstituents might contribute to the therapeutic potential of the *C. colocynthis* plant extract/fractions. Biologically active compounds were identified through GC-MS and HPLC analysis. The GC-MS analysis of the crude methanolic extract is presented in Table 2. Eighteen compounds were identified. The major compounds were ether, 3-butenyl pentyl, methyl stearate and 1-octadecenoic acid, methyl ester. A study by Kumar D et al., 2019, conducted a GC-MS analysis of *C. colocynthis* n-hexane and reported on methane, oxybisdichloro (82.55%), and trans-13-octadecenoic acid and methyl ester (4.38%). Previous studies of plant extract indicate the presence of various phytoconstituents, such as flavonoids and phenols [20]. The FT-IR analysis of the current study confirm the presence of hydroxyl (OH), cyclo alkane, phenol, C-O-C, C-H, C-S, and C-Br as functional groups in the range of wave numbers, given in Table 4. The functional groups confirm the HPLC phenolic and flavonoid results. The FT-IR analysis of the plant extracts depicted aromatics, carboxylic acids, nitro compounds, phenols, aldehydes, alkynes, alkyl halides, and alkanes.

The focus of the present study revealed that the fruits portion of *C. colocynthis* L. and analyzed whether the plant has hepatotoxic action against methanolic, n-hexane, and chloroform extract as hepatotoxins in Swiss albino rats to refute its claims in folklore medicine for liver issues. The extent of liver damage was determined by histological examination. Research by Vakiloddin and his team concluded that *C. colocynthis* fruits are beneficial and demonstrate hepatoprotective action, thus supporting that the plant extract is traditionally used to treat liver disorders [21].

The above-mentioned results indicate that the administrated dosage concentrations 150, 300, 450, and 600 mg/kg of (*n*-butanol, *n*-hexane, and chloroform) fractions to groups of rats under study did not show remarkable changes to liver cells. There is no ballooning and no fatty droplets, and Figure 1, Figure 2 and Figure 3 demonstrate no nuclear degeneration (A section). However, at higher concentrations, slight non-promising results were achieved by administration of dosage 600 mg/kg of (*n*-butanol, *n*-hexane, and chloroform). Fractions of methanolic extracts, slight ballooning, ECM accumulation, and nuclear variation were observed in histopathology examination; furthermore, the 600 mg/kg dose demonstrated intact architecture, with granular modifications in a few locations.

Research findings show that the stabilization of cell membranes, the regeneration of hepatic cells, and the normalization of serum parameters may be the mechanisms by which the 90% extract of *C. colocynthis* L. (200 mg/kg BW) provides in vivo hepatoprotective activity against paracetamol, so the extract is not hepatoxic at all, as shown in Figure 4, Figure 5 and Figure 6 [22]. The recent research result indicated that these are minimal changes, so the fractions at 600 mg/kg are not toxic at all and can be used for medicinal purposes because of the antioxidant compounds present in rich amount. Similarly, the research performed by [23] showed that treatment with *C. colocynthis* glycosides and alkaloids show significant improvement at various plasma and tissue levels. Treatment with *colocynthis* glycosides and alkaloids has hepatoprotective effects impact, instead of a hepatotoxic effect. The DPPH radical scavenging activity is a widely used method for determining the antioxidant capacity of any phytochemical extract. Among all the experimental plant extracts and methanolic fractions with the highest DPPH scavenging activity, the antioxidant assay on the concentration of 0.9 mg/mL of n-hexane fraction demonstrated significant DPPH scavenging activity (52.54 ± 0.49) with an IC_50_ value of 0.847687. The DPPH scavenging activity (52%) and, similarly, the minimum concentration 0.15 mg/mL of methanol extract give high DPPH scavenging activity (28%) with an IC_50_ value of 1.548036, followed by 0.15 mg/mL chloroform fraction (26%), n-hexane fractions (25%), and n-butanol fraction (17%). Thus, it was observed that, by increasing concentration of extracts and fractions from 0.15 mg/mL to 0.9 mg/mL, the percentage of DPPH scavenging activity will be increased. The same work was conducted by [24].

Another work performed by Reddy et al. [25], in which they indicated that hydrogen peroxide produces hydroxyl free radicals that can cause DNA damage, while *C. colocynthis* leaf extract showed the highest % inhibition of H_2_O_2_ 91.6% [26]. This indicates that many organic compounds or secondary metabolites are responsible for the antioxidant activity. In this study, methanolic fruit extract of *C. colocynthis* was screened, and this medical plant showed the highest anti-diabetic activity. Thus, this plant is used worldwide for the treatment of diabetic mellitus. The highest antioxidant and free radical scavenging ability of the fruit extract was observed at a concentration of 2500 mgmL^–1^ [27]. The thrombolytic effect with the butanol sample, clot lysis, was found at (7.5 ± 0.40), while with the chloroform test, clot lysis was observed at (3.4 ± 0.37). Furthermore, 250, 500, and 1000 g/mL *A. fragrantissima* (87.9 ± 1.0, 97.9 ± 5.1, and 112.5 ± 1.1 s, respectively), 500 g/mL and 1000 g/mL *C. colocynthis* (65.1 ± 1.0 and 106.4 ± 0.4 s, respectively), and 1000 g/mL *A. herba-alba* (15,703) had the highest PTT values. Using thrombosis tests, the anticoagulant impact of methanol extracts *C. colocynthis* was assessed, and the results indicate that the methanolic extract shows the highest thrombolytic activity. These findings suggest that these plants could be used to treat arterial and venous thrombosis [28]. As a consequence, a sample of anti-diabetic butanol was found. At 0.9 mg/mL, the highest amylase inhibition percent was reported to be 83.64 ± 0.27, followed by the methanol extract sample at 83.3 ± 40.47 mg/mL [29]. The 0.15 mg/mL sample percentage of amylase inhibition was estimated using hemolysis percent extract samples and methanolic fraction, which were determined to be 14.66 ± 0.22 of n-hexane at 0.9 mg/mL and 22.50 ± 0.43 of methanolic extract at 0.15 mg/mL, respectively. The plant materials used in extract production included roots, fruits, seeds, rinds, and leaves. The extracts were ethanolic, methanolic, or aqueous in nature, with daily doses varying from 10 to 500 mg/kg body weight. *C. colocynthis* is mentioned in all of these papers as a potential anti-diabetic medicinal plant. Total protein quantification was estimated for different fractions of fruit of *C. colocynthis* through Bradford method. The results revealed that crude extract contains higher protein concentration (600 µg/mL ± 0.010), as comparison to the concentrations, 50% and 70% concentration of ammonium sulphate precipitation (500 µg/mL ± 0.015), (400 µg/mL ± 0.003), respectively.

In a current SDS-PAGE study, the results revealed small-sized protein ranging from 100 to ~70 and ~55 kDa, ~40 to ~35 kDa, and ~15 to ~25 kDa in the 70%, 50%, and crude fractions. Different proteins were extracted from the seeds of the plant on the basis of MW and showed sizes of 150, 67, and 36 kDa. Hence, the current research can conclude from the previous and current study that there are dynamic proteins present in the whole and different parts of the fruit of *C. colocynthis*. In the docking process, all ligands were considered from PubChem. During the preparation of the binding site, the PDB format of the drug was selected using Pyrx for virtual screening. Molecular docking was used to evaluate binding affinities and understand the potential interactions between proteins and ligands. In the table above, binding energies are presented. Among the inhibitors, methyl stearate showed the highest binding energy. The ADME and drug-like properties of the molecules above indicate that they are moderately bioavailable in the gastrointestinal tract, but not permeable through the blood–brain barrier (BBB). The bioavailability radar considers six physicochemical properties of a drug to determine the molecule’s drug-likeness: saturation; Figure 2, boiled egg image of methyl stearate. The yellow regions indicates high brain penetration probability, while the white regions show high intestinal passive absorption permeability. The red dots indicate that the compound is not a P-Gp substrate. The polarity, flexibility, size, lipophilicity, and solubility in silico analysis of methyl stearate also showed that it may have antioxidative properties.

## 4. Materials and Methods

### 4.1. Plant Collection

Fruit samples (fully mature) were collected from the desert area (Hasalpur) of South Punjab, Pakistan in September–October 2021. The plant specimens were identified and authenticated by Dr. Muhammad Naeem (Taxonomist), Department of Botany, Government College University, Faisalabad, Pakistan (voucher specimen code, *C. colocynthis* (2734), University of Agriculture, Faisalabad, Pakistan).

### 4.2. Preparation of Plant Extract: C. colocynthis

Fruits of *C. colocynthis* were washed with distilled water and then treated with liquid nitrogen for crushing and drying, which were then pulverized with a pestle and mortar. An electric grinder was used to make a fine powder. The methanolic extract was made by dissolving 100 g of powder in 400 mL of 100% methanol and shaking it for 72 h at 37 °C on an orbital shaker. After the plant materials were settled, the supernatant solution containing plant compounds was collected and filtered. The crude methanolic extract was produced by rotational evaporation of the isolated and purified supernatant [30]. Furthermore, methanolic extract were distributed into fractions of n-hexane, chloroform, and n-butanol [31].

### 4.3. HPLC and GC-MS

The phenolics and flavonoids in methanolic extract of *C. colocynthis* fruits were quantified using high-performance liquid chromatography (HPLC). A CLC ODS C-18 with a diameter of 5.35 mm, 2.5 cm, and 4.6 mm was used as the column. Extract fractions of plant extracts with a concentration of 10 mg/mL were produced in their respective solvents. In addition to 20 mL of fractions, 20 mL of mobile phase A (H_2_O: acetoacetate 94:6, pH 2.27) and B (ACN 100%) have different conditions, varying from 15% B at 0 min to 45% B in 15–30 min to 100% B in 35–40 min. All samples were monitored at 280 nm with detectors that detect ultraviolet light [32].

### 4.4. GC-MS Analysis

A GC-MS was used to analyze the methanolic extract. This experiment was performed using a Clarus 580 chromatography equipment with a capillary column (5 percent phenyl, 95 percent methypolysiloxane) (30.0 MX 250 m) and a mass spectrometer (Polaris Q) (EI 70 eV). Helium at 1 mL/min was used as the carrier gas [33].The injection volume was 1L, and the split was 1/75. Temperatures for injection and monitoring were set at 250 and 280 °C, respectively. The temperature of the mixtures determining the temperatures of column were designed to rise at a rate of 11 °C/min from 50 °C to 200 °C, then 6 °C/min from 200 °C to 240 °C. The spectra of the major unknown compounds were compared to the spectrum of the known component in the NIST library.

### 4.5. FTIR-ATR Analysis

FTIR-ATR analysis of methanolic extract and methanolic fractions (n-hexane, chloroform, and n-butanol) for phytochemical screening. Methanolic extracts and fractions of *C. colocynthis* (n-hexane, chloroform, and n-butanol) were analyzed by Fourier transform infrared spectroscopy (FT-IR). By using a Perkin-Elmer spectrometer system, with a resolution of 4 cm^−1^ within a peak range of 450–4000 cm^−1^, the study validated bioactive compounds on the basis of functional groups identified in HPLC and GC-MS analyses [29,34].

### 4.6. Biological Activities

The biological activities performed inside the lab to check the potential of extract of *C. colocynthis*.

#### 4.6.1. Antioxidant Activity

Antioxidant potential of *C. colocynthis* was measured by following the method described. Antioxidant potential of *C. colocynthis* whole fruit hydro-methanolic extract was checked with DPPH radical by a micro assay with some modification. Volume of assay was adjusted to 100 μL. Totals of 90 μL of DPPH solution and 10 μL of different concentrations extract were placed in wells of a 96-well micro plate. DPPH with methanol was taken as negative control, while ascorbic acid was taken as positive control [35].

#### 4.6.2. Hemolytic Activity

The methodology employed to check the extracts’ hemolytic activity [36,37]. PBS served as the negative control, and 0.1% of Triton-x100 served as the positive control. The hemolytic potential of citrullus plant extracts and its fractions was measured via hemolytic assay escribed by [36]. Blood sample (3 mL) was taken in a falcon tube and washed three times with chilled phosphate buffer saline (5 mL) and centrifuged falcon tube contain blood at 2000 rpm for 10 min. A total of 20 μL of plant extract and its fractions was mixed with 2 mL of RBCs suspension in an Eppendorf and re-centrifuged for 10 min, and 100 μL of supernatant was diluted with chilled PBS (900 mL). PBS and Triton X-100 (0.1%) were used taken as negative and positive control, respectively. Measured the absorbance at 576 nm using 96-well micro plate.

#### 4.6.3. Thrombolytic Activity

Utilizing the methodology, as indicated by the clot, thrombolytic activity was carried out. Distilled water served as the negative control, and streptokinase served as the positive control. Previously described assay was performed to measure thrombolytic potential of citrullus plant [38]. Blood sample (2 mL) was collected and equally (500 µL/tube) transferred in four different pre-weighted eppendorf tubes. Tubes were incubated for 40 min at 37 °C, until blood clot is formed, after clot formation the serum was removed, then eppendorf tubes were re-weighted. Various concentrations of 100 μL of plant extract and each fraction were used. Sterile dist. water and streptokinase were taken as negative and positive controls, respectively. All the tubes were re-incubated for 90 min at 37 °C, and weight of lysis was calculated by following formula.
Clot weight = weight of clot containing tube-weight of tube
% lysis = (Weight of clot − weight of lysis) × 100

#### 4.6.4. Anti-Diabetic Activity

In-vitro antidiabetic activity was performed using α-glucosidase inhibitory assay, which was by following the method described by Ghauri, Ahmad, and Rehman, 2020, with little modification. A total of 100 µL was used for the total volume of the observed assay. Totals of 10 µL of extract (serial dilutions), α-glucosidase 10 µL (0.5 unit/mL), and 70 µL of 0.1 M phosphate buffer (pH 6.8) wells of 96-well plate were incubated for 15 min at 30 °C. A total of 10 µL of p-Nitrophenyl-α-D-glucopyranoside substrate solution was added and incubated for an additional 30 min. Test was performed three time [39].

#### 4.6.5. Anti-Bacterial Activity

Multiple bacterial strains were used to find the anti-bacterial activity of the sample by disc diffusion method, as explain below. Chosen bacterial list includes: *Pseudomonas aeruginosa*, *Klebsiella pneumoniae*, *Enterococcus faecalis*, and *Staphylococcus aureus*. For preparation of culture, nutrient broth was used. Nutrient broth was prepared by dissolving 13 g of nutrient broth in 1 L distilled water and mixing properly to form a yellow-colored solution. Autoclaved the broth and poured equally in 4 conical flasks and labelled accordingly by picking colony of selective bacterial strains by using sterile loop and transferring it into respective labelled flasks in sterilized laminar flow. Incubate the broth for 24 h at 37  °C and turbidity was observed, which indicates the bacterial growth [40].

For preparing agar plates to promote bacterial growth, 23 g of nutrient agar was dissolved in 1000 mL of distilled water in a conical flask and autoclaved. Transferred the agar in sterile petri plates in a laminar flow and left for solidification overnight. Witnessed for contamination, if any, and stored the plates for further testing.

##### Disc Diffusion Method

After solidification of agar, labelled the plates accordingly and poured 100 μL of bacterial culture on respective plates and, by using a glass spreader, spread the culture on agar plates. Sterilized discs were used, and 100 μL of crude and each precipitated samples, 50% and 70%, were put on the discs. Then, transferred all the three discs to the prepared plates and incubated for 24 h at 37 °C. Partially purified peptides having anti-bacterial activity showed visible zones of inhibition. Imipenem, vancomycin, and aztreonam were used as positive controls. Zones of inhibition were measured in mm by using zone reader [41].

#### 4.6.6. In-Vivo Hepatotoxicity of Fractions of Methanolic Extracts

##### Preparation of Sample for Hepatotoxicity

The Swiss albino tats were purchased from the University of Lahore, Lahore, Pakistan, and weighed 110–130 g. At the University of Central Punjab Animal House in Lahore, Pakistan, rats were maintained in an ideal environment.

##### Experimental Protocol

Table 9 shows that the 16 animals were divided into one control and three treated groups with methanolic fractions (n-butanol, chloroform, and n-hexane).

The control group dose is constant, and that is normal saline 1 mL/kg, but in treatment groups, each albino rat was treated with 150 mg/kg, 300 mg/kg, 450 mg/kg, and 600 mg/kg, respectively [31].

Selection and preparation of dose for the pharmacological screening:

The four doses of 150 mg/kg, 300 mg/kg, 450 mg/kg, and 600 mg/kg prepared according to animal body weight. According to following formula:$$\mathrm{Preparation}\;\mathrm{of}\;\mathrm{Dose}=\frac{\mathrm{Rat}\;\mathrm{wieght}}{100}\times \mathrm{Dose}$$

##### Histopathological Studies

All of the rats were put on the slaughter table under deep anesthesia, 48 h of liver preservation in a 10% formalin solution. Specimens were processed regularly, and sections were produced then stained with hematoxiline-eosine at a thickness of 5 microns (H&E). Qucik Lab Services (PVT) LTD examined the slides using light microscopy to assess the histological findings [42].

#### 4.6.7. Extraction of Peptide

##### Buffer Extract Preparation

About 72 g of the *C. colocynthis* fruit treated after liquid nitrogen was taken, then crushed in cold pestle and mortar in phosphate buffer saline with pH 7.4. It was stored at −20 °C. Subjected this sample to the centrifugation at 10,000 rpm for 10 to 15 min and separated the supernatant. Furthermore, refrigerated it at −20 °C for further use.

##### Salting In and Salting Out

The supernatant separated will now be further subjected for precipitation by treating with ammonium sulphate. Different concentrations of ammonium sulphate precipitation were used for salting, including 30%, 50%, and 70%. Dialysis membrane was used for the salting out procedure by pouring the sample in dialysis membrane and placing in magnetic stirrer for 12 h. After that, centrifuged the sample at 6000 rpm for 15 min, and pellet was saved in buffer for further use at −20 °C [43].

##### Bradford Assay

To measure the unknown sample, bovine serum albumin (BSA) was used as a standard at different concentrations to establish a standard curve for protein concentration. A total of 1 mg/mL stock solution of BSA was prepared, and dilutions of 10, 20, 40, 60, 80, and 100 µL were made to obtain the solution with a protein concentration ranging from 0–1000 µg/mL. Biotek ELISA reader was used to measure the absorbance by adding 250 µL of Bradford reagent and 10 µL of either BSA standard or unknown sample fractions in each well of 96-well ELISA plate. Absorbance was determined at 630 nm with each sample, and standard were run in triplicates. Standard curve was made depending upon the absorbance obtained at different concentrations of BSA standard. Total protein concentration of unknown sample was then calculated by comparing with the standard curve. The more the concentration of protein, the more intense the blue color [44].

#### 4.6.8. In Silico Studies of Antioxidant

##### Ligand Selection

From the GC–MS analysis, methanol crude extract *C. colocynthis*. The compounds were identified. Each compound was analyzed. A PubChem database was used to obtain the 3D structures of all the compounds.

##### Selection of Target Protein

A susceptibility protein was identified in the literature. This target protein’s 3D structure is available in the PDB database (https://www.rcsb.org) (Accessed on 12 October 2022). In the PDB database, Melatonin’s 3D structure was found, and its PDB ID is (6ME7) (5Z2f) (6WYz) (5Z2f) (5Z2f) (6WYz).

##### Docking Studies

PyRx software was used for docking studies for melatonin (6ME7), NADPH (5Z2f), Myloperoxidase (6WYz), and phytocompounds (ligands) of *C. colosynthis*. PyRx’s Open Babel option was used to prepare all the ligands and Discovery Studio 2021 to prepare the target protein. As part of the analysis, Discovery Studio 2021 was also consulted [45].

##### Pharmacokinetic Properties

Swiss ADME is a free tool for evaluating small molecule pharmacokinetics, drug-likeness, and medicinal chemistry properties. There are a bunch of free web-based tools for ADME and pharmacokinetics out there, such as PK-CSM4 and admetSAR5. Apart from unique access to proficient methods, such as boiled egg, Swiss ADME also includes strengths such as non-exhaustively, different input methods, and computation for multiple molecules, furthermore, displaying, saving, and sharing results either individually or globally. Finally, Swiss ADME is integrated into Swiss Drug Design [46].

##### Statistical Analysis

All the experiments in present research work were conducted thrice, and statistical analysis of the data was performed by analysis of variance [47,48].

## 5. Conclusions

The current studies inferred that *C. colocynthis* contained a good profile of biologically active compounds, i.e., pentyl, oxime-, methoxy-phenyl-Ethyl 4 chlorobutanimidoate), as confirmed through GCMS and HPLC analysis. Secondary metabolites derived from the active fraction depicted strong bioactivities, i.e., anti-diabetic activity, antioxidant activity, and hemolytic and anti-thrombolytic activity. Cytotoxic study of *C. colocynthis* extracts based secondary metabolites exhibited antioxidant potential, rather than cytotoxic effect, and no significant toxicity in rats’ livers was observed. The above-said study recommended that medicinal plants can be used to cure different diseases. Furthermore, the study indicates that plant metabolites have a significant role in the modern era and may be used in the future as an alternative source of antibiotic and vaccine agents to mimic the effect of resistance. The proteins extracted from *C. colocynthis* have potential against Gram-positive and Gram-negative bacteria. Thus, it may be concluded that the ripened fruit of *C. colocynthis* is a medicinally significant plant abundant with bioactive proteins and organic compounds. This plant can be used as a safe therapeutic drug, without any cytotoxic effect.

## Figures and Tables

**Figure 1 molecules-28-01743-f001:**
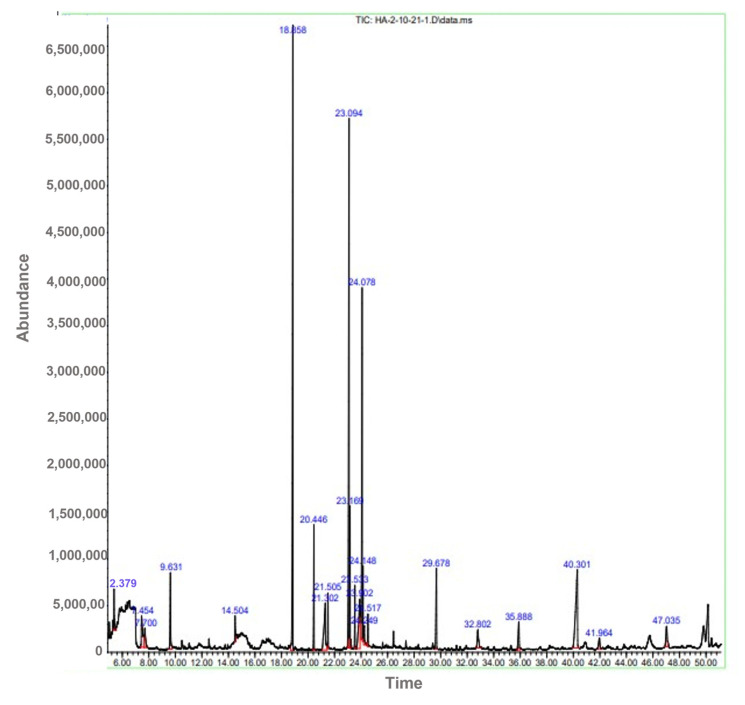
GC-MS Chromatogram of *C. colocynthis* fruit methanolic extract.

**Figure 2 molecules-28-01743-f002:**
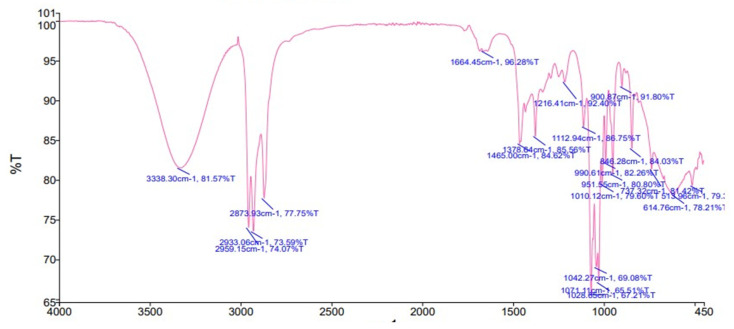
FT-IR spectrum of methanolic Extract of *C. colocynthis*.

**Figure 3 molecules-28-01743-f003:**
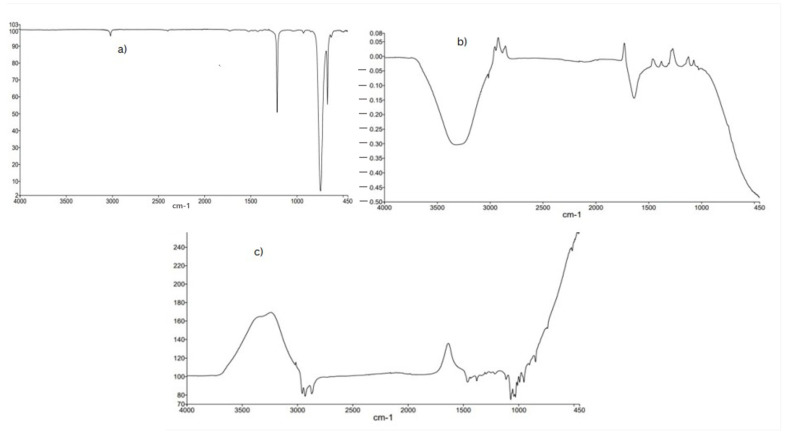
(**a**) The resulting peaks confirm the presence of wide range of functional groups of bioactive compounds. The results of FTIR spectrum of *C. colocynthis* confirmed the presence of alkane (C-H) with a peak at 250 cm^−1^, aromatic ester (C-O) with a peak at 1250 cm^−1^ and alcohols(O-H) at peak of 3363.67 cm^−1^. (**b**) FT-IR analysis confirms the presence of hydroxyl and carbonyl group in range of wave number (2500–3300 cm^−1^) and (1680–1755 cm^−1^), respectively. FTIR analysis in (**c**) shows Alcohols (OH) with a peak at 3700–3584 cm^−1^ and. Amines (NH) with a peak at 1650–1580 cm^−1^.

**Figure 4 molecules-28-01743-f004:**
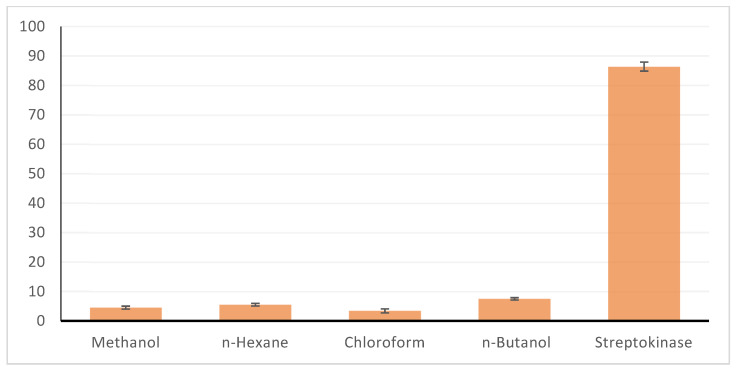
Thrombolytic activity of *Citrullus colonsythis* methanolic extracts and its fractions; the figure represents its results.

**Figure 5 molecules-28-01743-f005:**
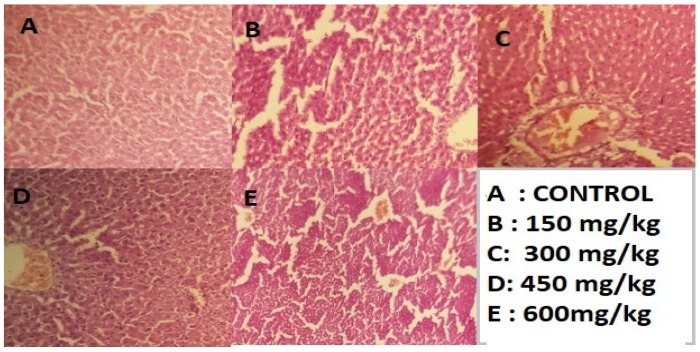
Hepatotoxicity histopathology investigations liver section given n-butanol (H & E 100X). Control (**A**); (**B**,**C**) Slightly ballooning of hepatocytes. No accumulation of ECM Overall liver tissue seems healthy, Fatty droplets were observed in the sinusoids. (**D**,**E**) Slightly ballooning, accumulation of ECM was observed, nuclear Variation, Vacuolar degeneration, and fatty droplets are present (100X).

**Figure 6 molecules-28-01743-f006:**
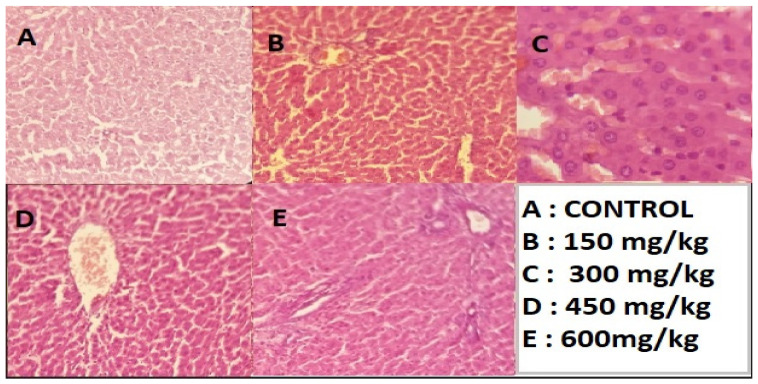
Hepatotoxicity histopathology investigations liver section given n-hexane Fraction (H & E 100X). Control (**A**); (**B**,**C**) Slight fibrosis with no ballooning, nuclear degeneration, no accumulation of immune cells. (**D**,**E**) Slight portal activity, Vacuolar degeneration, slight infiltration of macrophages nuclear variation, vacuolar degeneration, hydropic degeneration present (100X).

**Figure 7 molecules-28-01743-f007:**
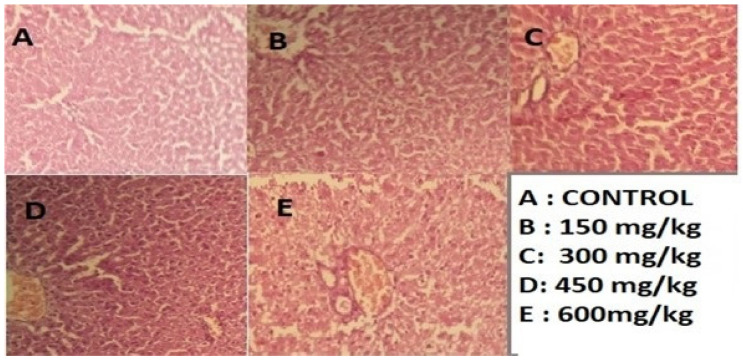
Hepatotoxicity histopathology investigations liver section given chloroform fraction (H & E 100X). Control (**A**); (**B**,**C**) No ballooning, Hydropic Degeneration, and nuclear variation are present, and No fibrosis or accumulation of immune cells. (**D**,**E**) Hepatocyte ballooning, very slight portal activation with some sort of hepatocyte damage, nuclear variation, vacuolar degeneration hydropic degeneration present (100X).

**Figure 8 molecules-28-01743-f008:**
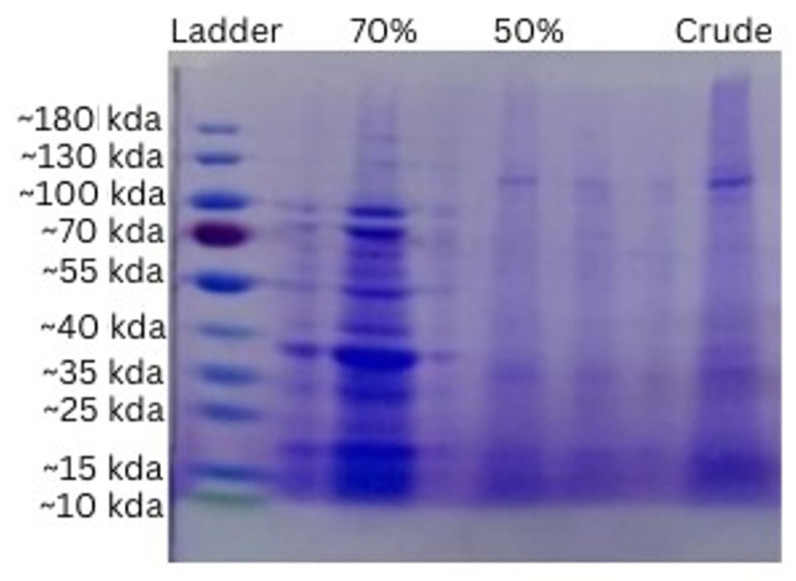
SDS-PAGE of different fractions of *C. colocynthis*.

**Figure 9 molecules-28-01743-f009:**
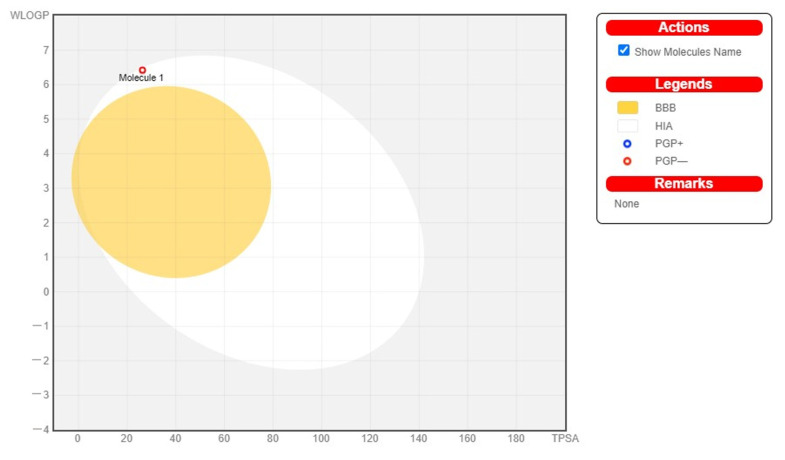
Boiled egg analysis of methyl stearate compound.

**Table 1 molecules-28-01743-t001:** Identifying bioactive molecules in GC-MS analysis of *C. colocynthis* fruit extract.

Sr. No.	Compounds	Retention Time	Area (%)
1	Ether, 3-butenyl pentyl	5.047	9.36
2	oxime-, methoxy phenyl	5.620	3.45
3	4-Dimethylbenzamide	6.186	0.94
4	Betaine	6.277	0.91
5	Ethyl 4-chlorobutanimidoate	7.417	1.82
6	Benzofuran, 2,3-dihydro	7.823	0.58
7	Cyclotrisiloxane, hexamethyl	8.604	1.26
8	Cyclohexasiloxane, dodecamethy	8.909	2.89
9	2-Methoxy-4-vinylphenol	9.745	5.27
10	p-Cymen-7-ol	10.059	0.51
11	3-Isopropoxy-1,1,1,7,7,7-hexamet.	12.616	2.13
12	2,4-Di-tert-butylphenol	13.471	0.88
13	Cyclooctasiloxane, hexadecamethyl	15.413	1.1
14	Triisopropyl phosphite	16.354	1.68
15	Cyclopropanecarboxylic acid, 2,2.	16.654	2.59
16	1-Octadecenoic acid, methyl est	16.863	6.44
17	cis-13-Octadecenoic acid, methyl	23.211	3.16
18	Methyl stearate	23.511	9.00

**Table 2 molecules-28-01743-t002:** HPLC analysis of methanol extracts of *Citrullus colocynthis*.

Sr. No.	Compound	Retention Time	Area (%)	Conc (ppm)
1	Quercetin	3.040	0.6	12.2127
2	Gallic Acid	4.807	5.0	74.8542
3	Vanillic Acid	13.287	4.7	122.6160
4	Chlorogenic Acid	15.513	3.4	110.7420
5	Syringic acid	16.967	2.4	25.2823
6	Feralic Acid	22.647	3.4	101.0458
7	Cinnamic acid	24.753	2.4	35.4156
8	Sinapic Acid	26.333	1.4	1.5497

**Table 3 molecules-28-01743-t003:** *C. colocynthis* methanolic extract containing FTIR identified functional groups with wave number (cm^−1^).

Sr. No.	Frequency Range (cm^−1^)	Functional Group
1	3400–3200	Hydroxyl Compound
2	2855–2975	Cyclo alkane
3	1458–1591	Phenol ring
4	1150–911	C-O-C
5	858–733	C-H
6	600–700	C-S
7	550–690	Halogen compound (bromo-compound) C-Br

**Table 4 molecules-28-01743-t004:** DPPH percentage scavenging (Antioxidant Activity) of extracts and fractions of CTC, values represent number of readings in each group = mean ± SEM.

Methanol Extract				n-Hexane Fraction		Chloroform Fraction		n-Butanol Fraction		Ascorbic Acid	
Sr. No.	Conc. (mg/mL)	DPPH % Scavenging	IC_50_	DPPH % Scavenging	IC_50_	DPPH % Scavenging	IC_50_	DPPH % Scavenging	IC_50_	DPPH % Scavenging	IC_50_
1	0.15	28.59 ± 0.45	1.54	25.66 ± 0.45	0.84	26.67 ± 0.45	2.87	17.48 ± 0.49	3	40.28 ± 0.28	0.29
2	0.3	29.38 ± 0.43		32.39 ± 0.49		28.57 ± 0.40		19.59 ± 0.51		50.58 ± 0.50	
3	0.45	29.36 ± 0.39		38.34 ± 0.31		29.61 ± 0.49		21.33 ± 0.56		60.57 ± 0.28	
4	0.6	30.57 ± 0.49		40.51 ± 0.49		29.39 ± 0.52		22.61 ± 0.44		70.59 ± 0.35	
5	0.75	38.41 ± 0.45		45.59 ± 0.51		30.44 ± 0.39		23.75 ± 0.27		80.63 ± 0.51	
6	0.9	40.57 ± 0.30		52.54 ± 0.49		34.59 ± 0.42		26.45 ± 0.49		93.50 ± 0.35	

The values were the average of triplicate samples (n = 3) ± S.D., two-way ANOVA shows (*p* ≤ 0.05), so results are significant.

**Table 5 molecules-28-01743-t005:** Anti-diabetic activity of methanolic extract and its Fractions of *Citrullus colonsythis*, values represent number of readings in each group = mean ± SEM.

Conc. (mg/mL)	Methanol Extract		n-Hexane Fraction		Chloroform Fraction		n-Butanol Fraction		Metformin (Standard)	
	Antidiabetic Activity	IC_50_	Antidiabetic Activity	IC_50_	Antidiabetic Activity	IC_50_	Antidiabetic Activity	IC_50_	Antidiabetic Activity	IC_50_
0.15	20.99 ± 0.24	0.56	15.99 ± 0.38	0.6	10.99 ± 0.99	0.51	25.99 ± 0.45		05.99 ± 0.10	2.3
0.3	25.60 ± 0.42		20.60 ± 0.34		15.60 ± 0.50		30.60 ± 0.24		10.66 ± 0.49	
0.45	35.19 ± 0.40		30.19 ± 0.37		25.19 ± 0.16		40.19 ± 0.20		20.19 ± 0.37	
0.6	40.71 ± 0.33		35.71 ± 0.47		30.71 ± 0.18		45.71 ± 0.45		25.71 ± 0.30	
0.75	75.23 ± 0.42		70.61 ± 0.23		65.23 ± 0.47		80.23 ± 0.41		60.23 ± 0.06	
0.9	83.34 ± 0.47		82.64 ± 0.36		81.54 ± 0.45		83.64 ± 0.27		78.67 ± 0.44	

The values were the average of triplicate samples (n = 3) ± S.D., two-way ANOVA shows (*p* ≤ 0.05), so results are significant.

**Table 6 molecules-28-01743-t006:** Hemolytic activity of methanolic extract and its fractions of *Citrullus colonsythis*, the values indicate number of readings in each extract class = mean ± SEM.

Methanol Extract			n-Hexane Fraction	Chloroform Fraction	n-Butanol Fraction	Triton X-100 Standard	
Sr. No.	Conc. (mg/mL)	Activity (%)	Activity (%)	Activity (%)	Activity (%)	Conc.	Activity (%)
1	0.15	2.50 ± 0.43	2.45 ± 0.40	1.49 ± 0.40	1.53 ± 0.30	0.1%	99.90 ± 0.15
2	0.3	2.30 ± 0.33	5.42 ± 0.45	2.67 ± 0.33	2.50 ± 0.40		
3	0.45	4.48 ± 0.33	8.43 ± 0.27	4.32 ± 0.06	3.32 ± 0.37		
4	0.6	5.45 ± 0.44	9.63 ± 0.23	5.19 ± 0.16	5.45 ± 0.49		
5	0.75	6.51 ± 0.47	11.50 ± 0.41	7.68 ± 0.22	8.37 ± 0.38		
6	0.9	8.43 ± 0.38	14.66 ± 0.22	9.53 ± 0.40	10.26 ± 0.40		

The values were the average of triplicate samples (n = 3) ± S.D., two-way ANOVA shows (*p* ≤ 0.05), so results are significant.

**Table 7 molecules-28-01743-t007:** Anti-Bacterial activity of different peptide fractions of *C. colocynthis*.

Sr. No.	Sample Fractions	Zone of Inhibition (mm)			
		*Staphylococcus aureu*	*Enterococcus faecalis*	*Klebsiella pneumoniae*	*Pseudomonas aeruginosa*
1	Crude	3.0 ± 0.001	2.76 ± 0.009	7.1 ± 0.178	-
2	50%	8.83 ± 0.022	10.12 ± 0.017	8.27 ± 0.001	6.4 ± 0.279
3	70%	6.96 ± 0.047	8.58 ± 0.012	5.86 ± 0.022	4.94 ± 0.473
4	Imipenem	19.4 ± 0.049	15.2 ± 0.057	18.1 ± 0.017	9.05 ± 0.042
5	Vancomycin	6.61 ± 0.062	-	-	-
6	Aztreonam	-	-	18.7 ± 0.206	-

The values were the average of triplicate samples (n = 3) ± S.D., two-way ANOVA shows (*p* ≤ 0.05), so results were significant.

**Table 8 molecules-28-01743-t008:** Molecular Docking of Bioactive Compound of Melatonin, NAD NADPH-Oxidase, and Myeloperoxidase.

Compound	PUB CHEM I.D.	Standard Melatonin	Myeloperoxidase	NADPH-Oxidase
Methyl stearate	8201	−6.4	−6.1	−5.1
Methyltrans-9-(2-butylcyclopentyl) nonanoate	14389759	−6.1	−6.2	−4.6
trans-11-Octadecenoic acid methyl ester	5364432	−6.1	−6.0	−5.3
9,12-Octadecadienoic acid	3931	−6.0	−6.1	−5.1
4,7-Octadecadiynoic acid methyl ester	569159	−6.0	−6.7	−5.2
Benzofuran, 2,3-dihydro	47756	−6.0	−6.2	−4.9
cis-13-Octadecenoic acid	5312441	−5.6	−5.7	−5.1
2-Butanone, 4-phenyl	91752799	−5.2	−5.4	−5.8
10.Benzamide, N-ethyl-N-(3-methylphenyl)-4-ethyl	533234	−5.3	−5.5	−4.2
Ether, 3-butenyl pentyl	537745	−5.0	−4.6	−4.2
TRIISOPROPYL PHOSPHITE	8304	−5.0	−4.9	−4.5
Cyclopropanecarboxylic acid	6451381	−4.8	−5.1	−4.5
2,4-Di-tert-butylphenol	7311	−4.6	−4.9	−4.1
[1,1′-Biphenyl]-2,5-diol	82722	−4.5	−4.6	−4.1
N,N-Dimethyl-p-(1-pyrrolyl)aniline	272429	−4.4	−4.4	−4.1
3,4-Dimethylbenzamide	21755	−4.2	−4.3	−3.9
ethyl 4-chlorobutanimidoate	13163358	−4.2	−4.3	−3.7
6-Chloropiperonyl alcohol	7015319	−4.2	−4.6	−3.9
2-Methoxy-4-vinylphenol	332	−4.2	−4.2	−3.9
Oxime-, methoxy-phenyl	9602988	−4.0730	−4.4	−4.1
Betaine	247	−3.7	−3.9	−3.6

**Table 9 molecules-28-01743-t009:** Characteristics of control and treatment groups of rats.

Doses	Group I (Control, Normal Saline mL/kg) n = 4	Group II (n-butanol Fraction mg/kg) n = 4	Group III (Chloroform Fraction vs. mg/kg n = 4	Group IV (n-hexane Fraction mg/kg) n = 4
1st	1	150	150	150
2nd	1	300	300	300
3rd	1	450	450	450
4th	1	600	600	600

## Data Availability

The data related to this research can be accessed upon a reasonable request at email: dr.afzal@ucp.edu.pk.

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
