# Peer review of "Characterization of Bioactive Compounds and Novel Proteins Derived from Promising Source Citrullus colocynthis along with In-Vitro and In-Vivo Activities"

_molecules, 2023, doi:10.3390/molecules28041743_

Round 1

Reviewer 1 Report

The article is an overview of to explore the biopotential of valuable biological active phytoconstituents and proteins from the medicinal plant C. colocynthis to cure various infections. Because bacterial resistance is going on increased day by day due to overuse and misuse of drugs so it’s a need of time that new source should be explore to find out secondary metabolites from medicinal plant to use as candidates drug molecule.

I appreciate to authors for their nice exhaustive work. I have suggestions to improve the manuscript:

-        Introduction need to rewrite, very short, how in introduction just 8 refrances.  

-       In material method section 2.1 it is not clearly mentioned about age of   the plant from which plant materials taken for extraction of oil. Age of the plant and season play important role for content. Please include the data in the section. 

-       GC chromatograms are missing in the manuscript. You add in the manuscript or give in supplementary file.

-       Fig.1 and Fig.2 both need replacement with clearly levelled one. Nothing visible in Fig.8 you replace with clear photograph.

-       Microorganisms and inhibition effectiveness: Authors need to summerize the Mode-of-Action in one table or colored figure, I think much well. Also this section need to increase due to this one from goal of article.

-       SDS-PAGE of different fractions of C. colocynthis this connected photo not one photo of gel. The authors add marker to determine the molecular weight of protein.

-       Where is statistical analysis??

-       Where is docking results?

-       Please check typo and Italic name in all article.

Author Response

Reviewer 1

I appreciate to authors for their nice exhaustive work. I have suggestions to improve the manuscript:

-        Introduction need to rewrite, very short, how in introduction just 8 refrances.

Response: Line 33-49: Dear reviewer, Introduction section have been revised and improved by adding new paragraph relevant to the current study, as well as new references have been cited.

-       In material method section 2.1 it is not clearly mentioned about age of   the plant from which plant materials taken for extraction of oil. Age of the plant and season play important role for content. Please include the data in the section. 

Response: Line 348: Dear reviewer, fully mature plant was collected and taken for the extraction method.

-       GC chromatograms are missing in the manuscript. You add in the manuscript or give in supplementary file.

     Response: GC chromatograms have been added in the supplementary data.

-       Fig.1 and Fig.2 both need replacement with clearly levelled one. Nothing visible in Fig.8 you replace with clear photograph.

Response: Dear reviewer, thank you for your valuable suggestion. We have revised the figures in the revised version of manuscript.

-       Microorganisms and inhibition effectiveness: Authors need to summerize the Mode-of-Action in one table or colored figure, I think much well. Also, this section needs to increase due to this one from goal of article.

Response: Line 204-209: Dear reviewer, section 3.9 has been revised and the table of all bacteria zone inhibition has already been added as Table no 7.

-       SDS-PAGE of different fractions of C. colocynthis this connected photo not one photo of gel. The authors add marker to determine the molecular weight of protein.

Response: A new image has been added as Figure 7 with ladder and sample bands.

-       Where is statistical analysis??

Response: Corrections has been incorporated as per reviewer comments. All the experiments in present research work were conducted in triplicates and statistical analysis of the data was performed by analysis of variance.

-       Where is docking results?

Response: The docking result are present in table 8.

-       Please check typo and Italic name in all article.

Response: Dear reviewer, thank you for your valuable suggestion. The manuscript has been thoroughly revised for English proofreading and grammatical mistakes. The correction has been made throughout the manuscript.

Reviewer 2 Report

suggestions in the manuscript

Author Response

Reviewer 2

Response: Dear reviewer, we would like to appreciate your kind suggestion and comments on our manuscript. We would like to thank you that, after addressing the comments from you and other reviewers, the manuscript has been thoroughly improved. Which is better for the readers noe. All of the changes suggested by you and other reviewers have been made possible with in the final manuscript and also response in the comments file.

Reviewer 3 Report

The paper "Characterization of bioactive compounds and novel proteins derived from promising source Citrullus colocynthis along with in-vitro and in-vivo activities" describes the antimicrobial effect of some extracts from the plant C. colocynthis against different bacterial specie. The article starts with a brief summary of the beneficial effect of this plant against other human patologies then focus on experiments that confirms the antimicrobial effects of some estracts.

The article is interesting from a scientific perspective but it requires to be revised since few syntax mistakes are present, along with the necessity to uniform the style. Following the problems that I detected. Don't be scared, except for the image they're just small and fast fixes:

line 159 - I suggest to add after "sodium dedcyl sulphate polyacrylamide gel electrophoresis" the known acronym between parentheses (SDS-page) to falicitate the lecture to younger and less experienced readers as well as scientists that are not familiar with it, overall since the acronym is used after this point. 

figure 7: definitely it cannot be presented like proposed. The markers are run differently than the sample, as clearly shown by the tracker.. the 10kDa marker is cut by the edge of the image. The sample gel is not well destained. Definitely is needed a new gel image with all the sample+marker without any graphical manipulation (only light/color fix could be allowed).

table 7: the table title is in a different page compared the table self.

line 258: after the citation [16] a fullstop is required.

line 261: there is a problem with the two sentences, capital letters and so on. Please check the line and change it in the correct form.

line 267-274: there is a problem with the uniformation of the text for %. I suggest to use in the whole paper the namer+% without space. I noticed that when u report a % sometimes the space is present and sotimes not.

line 277: Extract is in capital while it shouldn't be.

line 282: there a strange 2500mgmL-1thrombolytic... please check it.

ilne 283: the is a fullstop after the parentheses that is not necessary

line 284 and 286 both the plant name should be in Italic since they're latin plant name.

line 291 not sure of that but Butanol is right to be in capital? also not sure of the dash before amylase.

line 295 there is the number follwed by of without space at the beginning of the sentence.

to improve the manuscript I suggest to add in the line 258 number of the table you're talking about (table 8 I guess). 

line 320: maybe u can explain better the sentence between fullstops.

line 337: the coma at the end of the sentence

line 386: which broth do you use? it is all the same for every bacetial species?

line 390: please check the 37oC at the end of the line.

line 400: guess the come after 50 is a % instead.

line 433: 20 degrees Celsius could be changed in 20°C to uniform the text. The same at line 441. 

line 441: guess you missed the d of the past in save

line 445: the spaces after every coma.

line 447: ul should be change in the correct symble for micro

line 454: in silico should be corrected and set in Italic.

from 2.6.4.3 paragraph and for the whole page the right spaces after every paragraph are needed to uniform the paper and for a better visual order.

paragraphs 2.6.6 > please check the paragraph numeration from this point. I guess u skipped a 6.

line 485 there is an unwated fullstop

line 489 please check the possible double space after the fullstop.

bibliography: some articles have the capital letter at the beginning of every word, while other not. Please uniform the bibliography style.

Apart from the numerous small required corrections, the article is scientifically interesting and it should be published on Molecules after the fixes. 

Author Response

Reviewer -3

Comments and Suggestions for Authors

The paper "Characterization of bioactive compounds and novel proteins derived from promising source Citrullus colocynthis along with in-vitro and in-vivo activities" describes the antimicrobial effect of some extracts from the plant C. colocynthis against different bacterial specie. The article starts with a brief summary of the beneficial effect of this plant against other human patologies then focus on experiments that confirms the antimicrobial effects of some estracts. The article is interesting from a scientific perspective but it requires to be revised since few syntax mistakes are present, along with the necessity to uniform the style. Following the problems that I detected. Don't be scared, except for the image they're just small and fast fixes:

line 159 - I suggest to add after "sodium dedcyl sulphate polyacrylamide gel electrophoresis" the known acronym between parentheses (SDS-page) to falicitate the lecture to younger and less experienced readers as well as scientists that are not familiar with it, overall since the acronym is used after this point.

Response: Line 180: Dear reviewer, thank you for your valuable suggestion. The abbreviation has been corrected.

figure 7: definitely it cannot be presented like proposed. The markers are run differently than the sample, as clearly shown by the tracker.. the 10kDa marker is cut by the edge of the image. The sample gel is not well destained. Definitely is needed a new gel image with all the sample+marker without any graphical manipulation (only light/color fix could be allowed).

Response: Dear reviewer, we have run the gel again and replaced the figure 7 with a new better image.

table 7: the table title is in a different page compared the table self.

Response: Line 210, Table 7. The table has been moved to middle of the page now.

line 258: after the citation [16] a fullstop is required.

Response: Line 266: The mistake has been corrected.

line 261: there is a problem with the two sentences, capital letters and so on. Please check the line and change it in the correct form.

Response: Line 282: The sentence has been revised.

line 267-274: there is a problem with the uniformation of the text for %. I suggest to use in the whole paper the namer+% without space. I noticed that when u report a % sometimes the space is present and sotimes not.

Response: Line 293: The space between number and % has been removed.

line 277: Extract is in capital while it shouldn't be.

Response: Line 298: Corrected.

line 282: there a strange 2500mgmL-1thrombolytic... please check it.

Response: Line 303: It was a tying error. Mistake has been corrected.

line 283: the is a fullstop after the parentheses that is not necessary

Response: Line 304: Full stop has been removed.

line 284 and 286 both the plant name should be in Italic since they're latin plant name.

Response: Line 306-307: The plant name has been italicized.

line 291 not sure of that but Butanol is right to be in capital? also not sure of the dash before amylase.

Response: Line 311-312: Mistake has been corrected.

line 295 there is the number follwed by of without space at the beginning of the sentence.

Response: Line 315: space has been added.

to improve the manuscript I suggest to add in the line 258 number of the table you're talking about (table 8 I guess). 

Response: Line 279: The figure numbers have been provided.

line 320: maybe u can explain better the sentence between fullstops.

Response: Line 342-343: The sentence has been corrected.

line 337: the coma at the end of the sentence

Response: Line 359: The comma has been corrected.

line 386: which broth do you use? it is all the same for every bacetial species?

Response: Nutrient broth was used. It is not same for every bacterial species but the bacterial strains that we use mostly grow in this broth media.

line 390: please check the 37oC at the end of the line.

Response: Line 438: Corrected.

line 400: guess the come after 50 is a % instead.

Response: Line 446: Corrected.

line 433: 20 degrees Celsius could be changed in 20°C to uniform the text. The same at line 441. 

Response: Line 479: Corrected.

line 441: guess you missed the d of the past in save

Response: Line 487: Corrected.

line 445: the spaces after every coma.

Response: Line 491: Corrected.

line 447: ul should be change in the correct symble for micro

Response: Line 493: symbol has been corrected.

line 454: in silico should be corrected and set in Italic.

Response: Line 500: Corrected.

from 2.6.4.3 paragraph and for the whole page the right spaces after every paragraph are needed to uniform the paper and for a better visual order.

Response: Dear reviewer, thank you for your valuable suggestion. It was according to the guidelines of journal. After the 3rd subheading, there was no space.

paragraphs 2.6.6 > please check the paragraph numeration from this point. I guess u skipped a 6.

Response: Line 501: The numeration has been corrected.

line 485 there is an unwated fullstop

Response: Line 534: The full stop has been replaced with comma.

line 489 please check the possible double space after the fullstop.

Response: Line 538: The extra space has been deleted.

bibliography: some articles have the capital letter at the beginning of every word, while other not. Please uniform the bibliography style.

Response: We modified the bibliography style according to the MDPI style.

Apart from the numerous small required corrections, the article is scientifically interesting and it should be published on Molecules after the fixes

Response: Dear reviewer, thank you for your valuable suggestions and comments on the present manuscript.

Round 2

Reviewer 1 Report

 Accept in present form